# Waste Cooking Oils into High-Value Products: Where Is the Industry Going?

**DOI:** 10.3390/polym17070887

**Published:** 2025-03-26

**Authors:** Valentina Beghetto

**Affiliations:** 1Department of Molecular Sciences and Nanosystems, University Ca’ Foscari of Venice, Via Torino 155, 30172 Mestre, Italy; beghetto@unive.it or valentina.beghetto@crossing-srl.com; Tel.: +39-0412348928; 2Crossing S.r.l., Viale della Repubblica 193/b, 31100 Treviso, Italy; 3Consorzio Interuniversitario per le Reattività Chimiche e La Catalisi (CIRCC), Via C. Ulpiani 27, 70126 Bari, Italy

**Keywords:** waste cooking oil, industrial trends, recycling, polymer synthesis, circular economy

## Abstract

Waste cooking oils (WCOs) are generated globally in significant amounts by various sectors including hospitality, households, and industrial operations. Many nations currently lack dedicated legislation for managing WCOs, creating a pressing environmental challenge. At present, WCOs are primarily utilized in industries as raw materials for biodiesel production and energy generation. However, their role in second-generation biodiesel production is contingent on availability, often necessitating imports of either biodiesel or WCOs from other countries. The European Union has emphasized the importance of prioritizing biowaste for high-value alternative products beyond biodiesel to achieve carbon neutrality by 2050. Many reviews have been published in the literature reporting potential WCO applications to produce biolubricants, biosolvents, animal feed, asphalt additives, among others, however, no detailed analysis of industrial trends has ever been presented. Within this panorama, unlike existing reviews that focus on specific polymer classes derived from WCOs, this work sought to present a comprehensive industrial overview of the use of WCOs in creating high-value polymeric materials beyond fuel and energy, providing a general overview of patents published (or alive) in the last 10 years, together with the analysis of which innovative products are being introduced and sold on the market today.

## 1. Introduction

Over the past 15 years, the global demand and consumption of vegetable oils have grown significantly, rising from 83 million tons (Mt) in 2000 to over 217 Mt by 2023. Out of this total, approximately 167 Mt were utilized in biodiesel production, while the remainder was mainly consumed by the food, animal feed, and pharmaceutical industries [1,2]. Palm and soybean oils dominate the global vegetable oil market, accounting for 40.0% and 29.0% of the market share, respectively. Other oils, such as sunflower (9.7%), rapeseed (6.3%), peanut (4.1%), corn (2.1%), coconut (1.9%), olive (1.8%), and sesame oil (1.1%), are produced in smaller amounts [2]. Asia leads the global demand for vegetable oils, contributing over 54.0% of the total consumption, with Indonesia being the largest producer, followed by China, Malaysia, the USA, and Brazil [3,4,5,6,7,8,9].

Although biodiesel offers a promising alternative to partially replace fossil fuels, vegetable oils remain a vital source of nutrition, contributing to around 10.0% of the global average caloric intake, second only to cereals [10]. However, the increased demand for biodiesel, particularly in the USA, has been linked to the expansion of palm oil plantations in Southeast Asia, exacerbating issues such as deforestation and global warming [11]. These factors, along with high production costs and the threat to the food and feed supply chains, have reduced the appeal of edible oils as biodiesel feedstock [12,13,14].

As an alternative, waste cooking oils (WCOs), fats, and greases, along with emerging technologies like algae-based solutions, have gained attention for biodiesel production. WCOs are used cooking oils originating from the hospitality and industrial sectors that are no longer fit for consumption [1,15,16]. The improper disposal of WCOs can cause significant environmental harm such as odor generation, the pollution of soil and water bodies, and challenges in wastewater treatment [13]. These oils may also contain harmful oxidation or microbial by-products, posing risks to aquatic life and human health including gastrointestinal illnesses and even cancer [17,18,19,20,21,22,23].

The recycling and reuse of WCOs present a valuable opportunity to minimize the disposal costs and mitigate environmental damage. At present, the industrial application of WCOs is primarily focused on biofuel and energy production [24,25,26,27,28,29,30,31,32,33,34,35]. However, aligning with the EU’s goal of achieving carbon neutrality by 2050, the emphasis is shifting from using WCOs as fuels to their transformation into high-value chemicals [15,36,37]. This transition reflects the urgent need to reduce resource consumption and CO_2_ emissions by moving from a linear economy model of “take-make-dispose” to a circular economy driven by green chemistry and sustainable design [38,39,40,41]. Additionally, the growing crisis of the biodiesel market opens a window for the diffusion of value-added products from WCOs to help satisfy the Sustainable Development Goals of Agenda 2030 (ONU Agenda 2030) and the principles of green chemistry.

Many reviews have been published in the literature reporting potential WCO applications for the production of biolubricants [31,42,43,44], biosolvents [45], animal feed [46,47,48,49], asphalt additives [50,51], composite materials [52,53], polymers [54,55], fine chemicals [56,57,58], and non-aqueous gas sorbents [59,60], however, no detailed analysis of the industrial trends has ever been conducted. Within this panorama, unlike existing reviews that focused on specific polymer classes derived from WCOs, this work sought to present a broader analysis of recent industrial developments and applications with a general overview of patents published (or alive) in the last 10 years, together with the analysis of which innovative products are being introduced and sold on the market today. This review aims to provide a comprehensive industrial overview of the use of WCOs in creating high-value polymeric materials beyond fuel and energy.

## 2. Methods

This review was organized following several key steps (Figure 1): (1) selection of the database, publication type, subject areas, and keywords; (2) analyses of titles, abstracts, and conclusions of the selected patents and papers; (3) subdivision of the patents and papers in different subsections (for example, biosurfactants and biolubricants). This process allowed us to identify 20,000 patents that were analyzed, and from the preliminary analysis, only 100 patents were considered inherent and further taken into consideration for the present review.

Research papers and patents included in the review were collected through Web of Science, Scopus, Google Scholar, ScienceDirect, ResearchGate, European Commission portal [61], Statista [62], Eurostat [63] and World Bank [64], Orbit [65], EU Patent Office [66], Google patents [67], WIPO [68], and the U.S. patent office [69] and were published or alive between 2014 and 2024. Logical operators (AND, OR, NOT) were used in combination with different keywords such as “waste cooking oil”, “used cooking oil”, “biolubricant”, “biosurfactant”, “biosolvent”, “polyurethane”, “biopolymer”, “non isocyanate polymer”, “acrylic polymer”, “alkyd ester”, “epoxy resin”, “asphalt/construction additive, rejuvenating, anti-aging”, “circular economy”, and “ directives”. Specifically, one or a combination of two keywords were chosen for the selection of the papers and patents and only those published in the English language were reported. Other information on products available on the market were searched on Google using keywords such as “used” or “waste cooking oil commercially available products”, “buy WCO/UCO products”. Once we identified the relevant sources, references were reviewed to identify additional relevant information.

This review focused on the reuse of WCOs to produce polymeric materials and not on fuel or energy production. A further exclusion concerned the enzymatic degradation processes of WCOs to produce low-molecular-weight compounds.

## 3. WCO Production and Market

While extensive information is available on global vegetable oil (VO) production, obtaining precise data on the volume of waste cooking oils (WCOs) generated and available for recycling remains challenging. According to Teixeira et al., approximately 320 kg of WCOs are generated for every ton of oil used in cooking, with recovery rates ranging from 23% to 75% depending on the region [18]. Alarmingly, the analysis by Teixeira revealed that over 85% of the 23 regions studied lacked specific regulations for WCO management. Zhao et al. corroborated these findings in a more recent study, highlighting that progress in adopting sustainable WCO management practices has stagnated, with improper disposal into the environment still being the predominant alternative [70].

In the United States, data from the Environmental Protection Agency indicate that restaurants annually produce approximately 11.4 million tons (Mt) of WCOs, but only a fraction is managed responsibly [71]. The total WCO production in the U.S. is estimated at around 15 Mt [72]. Even in the European Union, which is considered progressive in waste management, significant gaps persist. Estimates suggest that the per capita production of WCOs is 8 L per year, translating to approximately 4 Mt annually for a population of 500 million. This amount is four times higher than the volume currently collected [43]. By considering the EU’s annual vegetable oil consumption (24 Mt) and Teixeira’s estimation that 32% is discarded, the annual WCO production could be even higher, potentially reaching 7.7 Mt [18]. These discrepancies align with the global disparity between vegetable oil consumption (220 Mt) and the roughly 50 Mt of WCOs collected worldwide [71], as shown in Figure 2.

Additional values on the amount of WCOs recycled were found in the literature and the data are summarized in Table 1 [73].

From these data, it is interesting to note that the “rest of the world” and Asia had the highest percentages of recycled WCOs, which are prevalently used for biodiesel production and exported to other countries [73].

At present, industries primarily utilize WCOs as feedstock for biodiesel and energy production [74,75,76,77,78,79,80,81,82]. Processes such as alkali-catalyzed transesterification, cracking, hydrocracking, pyrolysis, and gasification are used to convert WCOs into fatty acid methyl esters (FAME), a key component of biodiesel [34,54,83,84,85]. Furthermore, WCOs with varying viscosity and acidity levels have been employed in energy production through combustion [86,87].

While the use of WCOs can help address the high costs of biodiesel derived from virgin biomass and mitigate environmental impacts [88,89], their role as a primary feedstock for second-generation biodiesel production hinges on their large-scale availability. In many cases, local food waste is insufficient to meet the demand, necessitating the importation of either biodiesel or WCOs from other regions [90,91,92]. However, as highlighted by multiple life cycle assessment (LCA) studies, importing WCOs is not a sustainable solution for ensuring energy security, nor does it effectively contribute to reducing greenhouse gas emissions [73,93,94,95].

In this context, the recently approved EU Renewable Energy Directive [96] specifies that “the share of biofuels and bioliquids, as well as of biomass fuels consumed in transport, where produced from food and feed crops, shall be no more than one percentage point higher than the share of such fuels in the final consumption of energy in the transport sector in 2020 in that Member State, with a maximum of 7% of final consumption of energy in the transport sector in that Member State” [96]. Given this regulatory framework, prioritizing the use of WCOs for the development of high-value products over their use in biodiesel production is a more favorable approach.

Therefore, this review did not delve further into the biodiesel production from WCOs, focusing instead on alternative and more innovative uses for these materials.

## 4. WCO Regulatory Situation

Regarding the EU, efforts to regulate waste oils date back to the 1970s, beginning with Directive 75/439/EEC, which focused on the collection of used oils, minimizing the environmental risks, and encouraging the recovery and recycling technologies [43,97]. Over the past decade, the EU has introduced several regulatory revisions addressing the classification, storage, recovery, and disposal of WCOs originating from various sectors including hospitality, households, and industry. The European Commission Decision 2014/955/EU, amending Directive 2008/98/EC and 2000/532/EC, provides a harmonized waste classification list, which is periodically updated to classify different wastes [98]. However, it is important to note that the inclusion of a waste on this list does not automatically classify a material as waste; a substance is considered as waste only when it meets the definition outlined in Article 1(a) of Directive 75/442/EEC [99].

WCOs, categorized under European Waste Code (EWC) 200125, are not directly hazardous to human health, but improper disposal poses environmental risks. To address these concerns, Article 1 of Directive 2008/98/EC establishes measures aimed at protecting both the environment and public health by reducing the negative effects of improper waste management and enhancing the efficiency in reuse [100] Consequently, Member States are required to develop and implement waste management plans. However, balancing a unified EU policy with Member State sovereignty presents challenges, particularly in harmonizing waste management, recycling, and disposal strategies across borders [36,95].

By 2018, numerous Member States had yet to establish adequate waste management infrastructure. In response, Directive 2018/851 was introduced to amend Directive 2008/98/EC, aiming to prevent the overdevelopment of waste treatment facilities and optimize resource use [101]. This directive also aligns with the EU’s broader goal of reducing the dependency on raw material imports and advancing sustainable material management through a circular economy model. It establishes the legislative framework necessary to support the EU Action Plan and Sustainable Development Goals outlined in the 2030 Agenda, with the overarching goal of achieving carbon neutrality by 2050 [31,44,102,103]. A consistent and comprehensive legislative approach across the EU is essential to foster near-zero waste processes.

The End-of-Waste Directive 2008/98 represents a key milestone in this regard, defining criteria for converting waste into valuable secondary raw materials [101,104]. According to Directive 2008/98/EC, “recovery” refers to processes where waste is used to replace virgin materials that would otherwise be used, either within industrial operations or more broadly in the economy [104]. Meanwhile, “recycling” is described as any recovery operation in which waste materials are processed into new products, materials, or substances for their original or alternative purposes excluding energy recovery or backfilling operations. To achieve “End-of-Waste” status, materials must meet specific conditions: they must be useful for a particular purpose, they must have an existing market or demand, they must meet applicable legislative and technical standards, and their use must not result in negative environmental or human health impacts [104].

In the United States, regulatory measures regarding waste oils have also evolved over time. In October 1997, the Environmental Protection Agency (EPA) declined a petition from several agricultural trade organizations requesting relaxed spill response regulations for facilities storing vegetable oils and animal fats under the Facility Response Plan (FRP) rule [104]. Subsequently, in 2000, the EPA revised the FRP rule to clarify its applicability to facilities that manage, store, or transport significant quantities of animal fats and vegetable oils, particularly those handling large-scale transfers over water or storing at least one million gallons of oil. This revision was aligned with the Edible Oil Regulatory Reform Act, distinguishing animal fats and vegetable oils from other oils based on their properties and environmental impacts [105].

The updated rule introduced a methodology for estimating worst-case discharge planning volumes for these oils, using principles similar to those applied to petroleum oils while incorporating tailored factors for recovery estimation in water and on land. Additionally, separate regulatory sections were designated for animal fats and vegetable oils while maintaining the original response planning categories for small, medium, and worst-case discharges. The rule also introduced new definitions and a classification system dividing oils into Groups A, B, and C according to their specific gravity.

Further regulatory changes occurred in 2002 with the revision of the Spill Prevention, Control, and Countermeasure (SPCC) rule, which was updated to reflect the requirements of the Edible Oil Regulatory Reform Act [106]. The revised rule was structured into subparts: Subpart A outlined general requirements for all facilities, Subpart B addressed petroleum and non-petroleum oils (excluding animal fats and vegetable oils), and Subpart C detailed regulations for animal fats, greases, fish and marine mammal oils, and vegetable oils derived from seeds, nuts, fruits, and kernels. This revision also highlighted the environmental impact of oil spills on soil.

To align with the Circular Economy Action Plan, the establishment of a comprehensive legislative framework is essential for advancing sustainable practices, encouraging waste upcycling, and fostering near-zero waste processes [31,43,44].

The Clean Water Act also plays a crucial role in managing waste disposal, including WCO, to prevent pollution [107].

However, recent political trends have seen a relaxation of certain environmental regulations, leading to concerns about increased waste and pollution. Despite this, many states have implemented their own regulations and incentives for biodiesel production from WCOs, promoting a circular economy approach [108].

China has seen a significant increase in the use of WCOs for biodiesel production, driven by government policies aimed at reducing waste and promoting renewable energy. The National Development and Reform Commission (NDRC) and the Ministry of Ecology and Environment (MEE) oversee regulations regarding waste management and environmental protection. Very recently, the Chinese State Council issued guidelines for expediting the establishment of a waste recycling system [109]. According to the document, the construction of waste recycling systems is considered crucial to ensure the security of national resources, actively and steadily promoting carbon neutrality, and accelerating the green transformation. It established that by 2025, a waste recycling system covering all fields and links will be initially set, and by 2030, a comprehensive, efficient, regulated, and well-organized waste recycling system will be built up to valorize various wastes and push the overall level of waste recycling to the top ranking in the world.

Efforts should also be made to improve the level of recycling and reuse of waste to strengthen the comprehensive utilization of bulk solid waste and enhance the efficient use of renewable resources.

Meanwhile, the resource recycling industry will be fostered and improved according to the guidelines [109]. However, challenges remain including illegal disposal practices and a lack of stringent enforcement. The government’s recent focus on economic growth may overshadow environmental concerns, impacting WCO initiatives.

India is experiencing a growing interest in converting WCOs, mainly focusing on biodiesel production, which is supported by the National Biofuel Policy [110]. The regulatory framework is still evolving, with the Ministry of Petroleum and Natural Gas leading the efforts to promote the use of biodiesels. As an example, in December 2023, the government of India started a campaign to promote WCOs for the purposes of biodiesel production [111], and state-owned oil companies have pledged to buy biodiesel made from UCOs over the next three years. These goals aim to mitigate the health hazards, environmental destruction, and reduce the cost of oil imports. However, the enforcement of regulations can be inconsistent, and the market is often hindered by a lack of infrastructure for collection and processing. Political factors, including economic priorities, can lead to insufficient attention to environmental issues related to WCOs.

## 5. Where Is the Industry Going?

Currently, thousands of companies over the EU are entitled to collect WCOs from households, hospitality, and industrial sectors, nevertheless the EU currently imports 80% of the WCOs used for biodiesel production, mainly from China (60%) [112]. Although the global airline industry is pushing for WCOs to become a key ingredient in sustainable aviation fuels, general skepticism has arisen from biodiesel producers asking for greater transparency to avoid WCOs becoming a backdoor for the illicit use of palm oil. In fact, the high demand for WCOs has raised the risk of fraud, where VOs like palm oil are suspected of being mislabeled as ‘used’, taking advantage of the higher price of green fuels. Several countries together with the EU Commission have presently launched investigations into fraudulent Indonesian biodiesel, potentially transiting through China and the UK to circumvent taxes [113].

To make things worse, it should be further considered that although the use of WCOs can reverse the noncompetitive price of biodiesel from virgin biomass and reduce the negative environmental impacts [88,89], the use of WCOs as a primary feedstock for second-generation biodiesel production depends on its large-scale availability, and since local food waste is often insufficient, the import of biodiesel or WCOs from other countries is required [90,91]. As evidenced by various LCA studies, the import of WCOs is not a sustainable practice to guarantee energy supply, nor does it contribute to reducing the gas emissions [73,91]. Additionally, the use of WCOs as feedstock for the biorefinery is struggling to find a place in the market, which is highly depressed in favor of the biodiesel and energy sector, which nonetheless is passing through a very critical moment in the EU. In fact, due to the strong reduction in biodiesel selling prices as a consequence of competition from Chinese companies, in 2024, Chevron closed its biodiesel plant in Germany, BP scaled back plans for sustainable aviation fuel and renewable diesel in the EU, and Shell halted the construction of a big biodiesel plant in Rotterdam [114]. Although the European Commission announced that provisional import duties will be imposed on Chinese biodiesel, competition for biodiesel production demonstrates that higher value-added products from WCOs, prepared with innovative technologies, are the answer to support EU market competitiveness, moving away from the biodiesel market.

That being said, most of WCOs recycled in the EU are presently employed for biodiesel and energy production and very few of the many technologies described in the first part of this work are exploited industrially (Figure 3).

This is of course strictly connected to market demand and economical sustainability, with a consequent reduction in more innovative but costly alternatives to WCO valorization. In fact, excluding the fuel and energy market, other products such as biobased lubricants, surfactants, fertilizers, soaps, and additives for cosmetics and animal feed available on the market are mostly produced from VOs (Table 2) [115], while those from WCOs remain a niche market. A more detailed analysis on the industrial maturity of different VO versus WCO applications is reported below.

Different biolubricants (BLs) that are commercially available on the market are produced by esterification or transesterification from VOs and are commercialized by Exxon Mobil (US), Royal Dutch (Netherlands), Total S.A. (France) Cargill Inc. (US), BP (UK), FUCHS Group (Germany), and Panolin AG (Switzerland). Although the BL market reached USD 2.95 billion in 2024 and is expected to grow at a compound annual rate of growth (CAGR) of 13.7% from 2025 to 2030 [116], the price of BL is 30–40% higher than fossil-based lubricants, inevitably impacting the market growth perspectives. Additionally, from a thorough analysis of the information available on the websites of BLs and from the French agro-biobased database [117], it was not possible to identify any BL available on the market produced from WCOs. This finding is consistent with reports from the biolubricant industry, which is seeking alternative raw materials to VOs in synthetic esters and polyethylene glycols while WCOs are not even included on the list (Marker Research).

Recently, King Abdullah University of Science and Technology (Saudi Arabia) was granted a patent to produce BLs from triglycerides and free fatty acids (FFAs) derived from WCOs and epoxidized triglyceride (US11767484, granted alive) [118]. Cargill also recently filed a patent (WO2024206034 A1, pending alive) on the production of BL, with an easy and economically sustainable protocol foreseeing the self-condensation of hydroxy fatty acids [119,120]. This technology appears promising in terms of opening up the market of BLs from WCOs as they can be easily integrated by the biodiesel industry, and moreover, all components employed are bio-based.

As far as biosurfactants (BSs) from WCOs are concerned, a large gap separates the research from industrial use and commercialization. In fact, the BS enzymatic processes reported in the literature achieve rather low yields in very long times, thus WCO-derived BSs are generally not economically competitive compared with fossil-based surfactants.

For example, Cavalcanti and coworkers [121] reported the scale-up production in a 50 L reactor of rhamnolipid biosurfactant in the presence of Pseudomonas aeruginosa reaching, in the best reaction conditions, 2.0 g/L/h production after 7 days of fermentation. To the best of the author’s knowledge, Amphistar is the first start-up company selling BSs produced from agrofood waste, among which WCOs, employing a highly productive yeast Starmerella bombicola, produce over 25 glycolipid biosurfactants. The technology has been patented and scaled up to a 15 m^3^ scale to deliver high-purity products that are ideal for a wide range of applications [122,123].

Concerning polymer additives, various products exist derived from VOs, such as Cargill Vikoflex^®^ 9010, a plasticizer, stabilizer, and acid scavenger, or Palsgaard’s Einar^®^ additives, but equivalent products derived from WCOs are apparently not commercialized to date. In 2013, Arkema Hydrogen Peroxide and Jiangnan University patented an invention relating to a method for preparing an environmentally friendly plasticizer by the epoxidation of WCOs with high concentration hydrogen peroxide (60–70%) as the oxygen source and formic acid as the catalyst [124]. The invention is now lapsed dead, probably because of the high production costs and low competitivity of the products patented.

As far as the development of epoxidized soybean oil for polyvinyl chloride additives is concerned, the industry has made a great effort not only to optimize synthetic protocols [125,126], but also to implement the efficiency of reactors, filtering systems, and other technological solutions for their economically sustainable production [127,128,129].

With regard to polyols and epoxides, the manufacturing from VOs has been commercialized by many companies such as Polylabs, Evonik Industries AG, The Dow Chemical Company, BASF SE, DuPont, Cargill Inc., Mitsui Chemicals, Hairma Technology (Nantong, China), Sirui Advanced Materials, Shandong Haobo Biomaterials, as testified by the numerous patents filed on the argument (US20220153654A1, CN114853972B, and Appendix A) [119,130]. Some WCO-derived polyols are gradually being introduced into the market, for example, those by Hairma (HM-10100R) or Whchem (Wanol^®^) and Polylabs, to replace epoxidized soybean oil for polyurethane prepolymer synthesis and to substitute part of the polyether polyols in the production of different density foams for running shoes and vehicle seats as well as additives for other industrial applications. GNO Chemicals (Awishkar Group) patented a method for manufacturing polyurethane prepared by a multistep protocol to produce polypropylene glycol diisocyanate polyurethane [131].

Interestingly, quite a number of different patents exist on the production of polyols from WCOs and their application for polyurethane production, for example, US8501826 B2 of the Malaysian Palm Oil Board [132], CN107722344B by L. Bin and L. Liping [133], CN102504190B by Nantong Haierma Technology [134], and CN103436367A by the Jiangsu Rebo New Material Technology Company [135]. Additionally polyurethane-amide adhesives using glycerol and fatty acid obtained from VOs or WCOs and poly methylene bisphenyl isocyanate employed as adhesives for environmentally friendly thermosetting materials have been patented [136]. Nonetheless, this rather unique patent of Konkuk University Industrial Cooperation is now lapsed dead, and apparently the adhesives never reached the market.

Concerning non-isocyanate polyurethanes (NIPUs), although no commercial product is available on the market, the interest of the industry appears clear from an analysis of patents filed in the past years, two of which comprise the use of VO linseed oil epoxidized polyols for the synthesis of NIPU (CN115490851B, EP4149999A4) [137,138]. Interestingly, these patents follow a similar synthetic strategy to the ones reported in the literature [139,140].

With regard to the production of alkyd resins, it is evident that all commercialized products found were produced from VOs and none from WCOs. Still, different patents report the synthesis of these polymers from WCOs, for example, the Changzhou Guanghui Chemical Inc. 2018 patent (CN109096475B, granted alive) [141] regarding a method for the preparation of alkyd resins from a WCO alcohol acid prepolymer in combination with methylpropanediol, 1,2,3-trihydroxypropane, and isophthalic acid. In 2019, the Hanilcon Corporation and Chaoyang Industry also patented a method to produce elastic pavements in which conventional chemical binders, such as styrene and the like, are replaced with an oil-modified alkyd resin manufactured from WCOs, with high compressive strength, freezing/melting resistance, chemical resistance, and is devoid of any harmful substances. The formulation is said to be adequate for paving sidewalks, bike paths, and walking trails, among others (KR102150710B1, granted alive) [142]. The patent is possibly bound to be on the market in the future since the Hanilcon Corporation and Chaoyang Industry are leaders in the Chinese tire market with an annual revenue of 3.9 M€.

The U.S. patent US8895689 B2, owned by Valspar Sourcing, Inc., Minneapolis, MN (USA) (earliest priority date 2006, granted alive), describes a method, consistent with methodologies known from the literature, for producing alkyd resins using WCOs containing at least 4% unsaturated fatty acids, alongside polyols and polycarboxylic acids, which react to form alkyd resins [143]. Regarding the panorama of WCO applications for asphalt mixtures, there are very few examples of industrial-scale applications [144]. In particular, Jain and Chandrappa reported one of the very rare examples of reclaimed asphalt pavement (RAP)/WCO mixtures used to pave different stretches of a road in Alabama as well as a comparison of their performances to standard asphalt mixtures after 24 months. Although the RAP/WCO pavement showed a lower resistance (higher density of longitudinal and transverse cracks), nonetheless, according to the authors, the protocol reported is very easy, potentially applicable at large scale, and economical studies further evidenced a significant reduction in the construction costs when RAP/WCOs were employed, with up to 34–40% of savings.

Checking the availability of the biobased asphalt additives indicated that just one producer could provide them, and moreover, they were not from WCOs. In fact, DERNATAC^®^ P105, patented by DRT—France [145], is a fully biobased resin produced from Tall-Oil rosin, a by-product of paper mills, esterified with pentaerythritol. This product provides properties that are interesting in the coating, varnishes, or adhesives industries, especially as additives in green bitumen. It is used to provide a high quality equivalent to VO-based binders for asphalt used in insulating coatings or roads.

The high level of industrial interest on the use of WCOs for polyurethane (PU) production is testified by the number of patents, as summarized below:

Oilstone, a South Korean company primarily engaged in the sale of asphalt additives and lubricants, has various granted and pending patents for the production of asphalt concrete recycling additives containing between 15 and 25 wt% of WCOs [146,147], although no specific product from WCOs is advertised on the company website (Oilstone.net). Hwashin and Road Seal, also important Korean leaders in road construction (Hwashinbolt UK), patented two applications for the use of WCOs as aggregate binders [148] and asphalt sealants [149], but the first has expired and the second has been invalidated, and currently no product is available on the company website.

The Hubei University of Arts and Science recently filed a patent, still pending, for the preparation of asphalt regenerants comprising WCOs, a surfactant, and a reinforcing agent. The low viscosity asphalt regenerant is said to have excellent wettability and permeability, allowing RAP recycling for pavement applications [150]. Another Chinese university, Fuzhou University, patented an application in 2024 for regenerating aged asphalt based with WCOs, esterified by methanol in the presence of a sodium hydroxide solution, followed by epoxidation with hydrogen peroxide, glacial acetic acid, and concentrated sulfuric acid to obtain regenerated asphalt. The recycled asphalt was employed to regenerate RAP with good temperature performance, rheological property, and shear resistance, proving adequate for road construction engineering [151]. Additionally, Chongqing University and the Chongqing Pavement Technology Company have had a granted patent since 2016 for desulfurizing waste tire powder with WCOs. The method foresees mixing powdered waste tires with WCOs at temperatures between 220 and 280 °C. The desulfurized waste tire powder is then used as an additive for RAP recycling [152]. Although these technologies are not presently industrially exploited, they are evidence of a potentiality that may hopefully turn into business in the future.

A case of success is represented by the Lyondell and Nestle partnership, who launched a new plant for the production of a variety of polyolefins produced from CirculenRenew products sourced from biobased wastes and WCOs, such as CirculenRenew, used for food packaging and/or high-quality requirement films, such as surface protection films, although no patent has yet been made public on the technology employed for their production (LyondellBasell). It is possible that the technology employed may be similar to the one adopted by the Orlen Unipetrol Group in its new plant in the Czech Republic, which was based on the replacement of crude oils with hydrogenated vegetable oils (HVOs) derived from WCOs for the production of biocomponents for fuels, monomers, and polymers. This technology combines the possibility to produce both biodiesel and polymers and is based on consolidated hydrogenation and purification steps that are well-known by the industry and is therefore a very appealing solution for the industry (Orlen).

Other patented solutions regarding the use of WCOs as rubber plasticizers (AU2020102505 A4, granted alive) [153] and dispersants for wood plastic composites (CN103611468A) are interesting potentials for future industrial products.

A less auspicious outlook appears to be the rather large number of technologies patented by various international institutions for innovative applications of WCO recycling, many of which expired very shortly after their presentation. For example, CN109266026A, disclosing the use of an oxidized graphene and WCO composite modified warm mix asphalt (earliest priority date 2018, revoked dead), or CN113845848 on the use of WCOs for high-flexibility pressure-sensitive adhesives for 3D and 4D printing (earliest priority date 2021, lapsed dead) [154,155].

Other patents on apparently more industrially sustainable technologies, such as additives and rejuvenators from WCOs, were also lapsed dead or revoked very shortly after their application such as CN117511240A (earliest priority date 2018, lapsed dead), CN109082132A (earliest priority date 2018, lapsed dead), or CN109439006A (earliest priority date 2018, revoked dead). Less common applications have also been patented on the use of WCOs as coating materials for slow/controlled release fertilizers (CN103755866B), or wax materials (CN114854449A) but not with greater success, as the patents lapsed dead shortly after their deposit [156,157,158,159,160]. Further information is reported in the Appendix A.

## 6. Conclusions

In conclusion, this is the first paper to report a general overview and comparison of the latest technologies implemented by both the scientific community and the industry for the upcycling of WCOs for the production of polymers. From this analysis, a general coherent trend emerged between innovative solutions studied by the scientific community and those adopted by the industry. Nonetheless, since most of the companies selling WCO-derived polymers are primarily biodiesel producers, industry privileges the adoption of technological solutions that can be easily implemented in biodiesel production plants. Interestingly, market availability is also significantly affected by legislative restrictions, so bioplasticizers and biopolyols are increasingly available on the market as alternative environmentally sustainable solutions to phthalates and fossil-based or VO-derived polyols. Additionally, bioasphalt additives are gaining interest, probably due to the high potentiality of this market, which is expected to reach 183 Mt by 2027 [161]. Interestingly, more disruptive and expensive technologies for the production of BSs have also found a collocation on the market, thanks to the initiative of smaller startup companies (Amphistar).

The data reported in this study further highlight that, although from a technological point of view many different strategies and applications exist for WCO upcycling, only very few have been implemented at the industrial level. This clearly testifies that further efforts need to be undertaken by both the scientific community and industry to reduce the process complexity. It remains that the complexity and variability of the WCO composition, together with an inadequate collecting system, are among the main barriers for the introduction of WCOs as feedstock at the industrial level, and their competing use for biodiesel production is a drawback, reducing their availability. To overcome these problems, stronger legislative strategies should be adopted in order to harmonize the collecting system throughout the EU and promote awareness campaigns among consumers on the importance of the separate collection of WCOs and their potentiality for the production of high-value-added products. Interestingly, today, unlike in the past, the growing crisis of the biodiesel market opens a window for the diffusion of value-added products from WCOs to help satisfy the Sustainable Development Goals of Agenda 2030 (ONU Agenda 2030) and the principles of green chemistry.

## Figures and Tables

**Figure 1 polymers-17-00887-f001:**
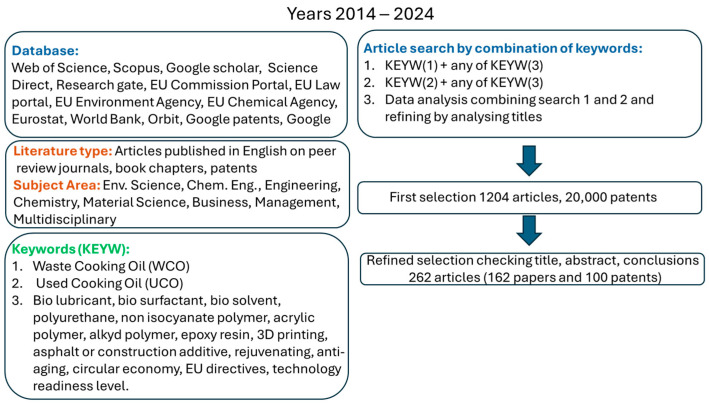
Methodology for the selection of the reviewed papers and patents.

**Figure 2 polymers-17-00887-f002:**
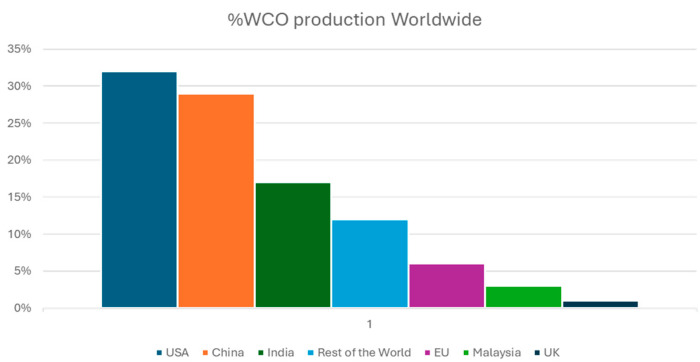
Geographic distribution of global WCO production [71].

**Figure 3 polymers-17-00887-f003:**
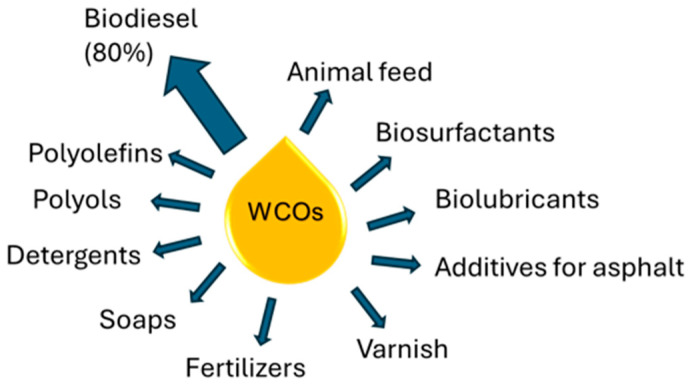
Main use of WCOs worldwide.

**Table 1 polymers-17-00887-t001:** Annual production of WCO and recycled WCO by geographical region.

N°	Region	Annual WCO Produced		Annual Mt WCO Recycled	
		Mt	(%) ^(a)^	Mt	(%)
1	USA	15.0	32	2.7	18
2	Asia (China, India, Malaysia)	24.5	49	7.6	31
3	EU	4.0	6	1.0	25
4	Rest of the world	6.5	13	2.8	43
Total		50.0	100	14.1	28 ^(b)^

^(a)^ Calculated as the percentage of a region Mt production divided by the total WCO Mt yearly production worldwide (50 Mt). ^(b)^ Calculated as the percentage of Mt total recycled WCOs (14.1 Mt) divided by the total WCO Mt yearly production worldwide (50 Mt).

**Table 2 polymers-17-00887-t002:** Commercially available products from VOs and WCOs.

Company Name	Type of Product	Source	Link
Adm	Additives for asphalt	VO	https://www.adm.com/ (Accessed 14 January 2025)
Biosynthetic technologies	Biolubricants, solvents, additives for cosmetics	VO	https://biosynthetic.com/ (Accessed 15 January 2025)
Cargill	Biolubricant (Agri-pure™ AP-406), polyols, polymer additive (Vikoflex^®^ 9010)	VO/Soy Oil	https://www.cargill.com/ (Accessed 18 January 2025)
Gamblin Colors	Soy-based alkyd resin for paints	VO	https://gamblincolors.com/ (Accessed 15 January 2025)
Palsgaard	Polymer additive (Einar^®^)	VO	https://www.palsgaard.com/en/ (Accessed 19 January 2025)
Renewable Blue	Biolubricants (Bio-SynXtra™)	VO	https://renewablelube.com/ (Accessed 20 January 2025)
Repsol	Biolubricants (MAKER BIO CHAIN 68), bioplastics (PHA), cosmetic creams, detergent, paint, varnish	VO	https://www.repsol.com/en/ (Accessed 20 January 2025)
Vanair Inc.	Biolubricants (Vanguard Green oil)	VO	https://vanair.com/vanguard-oil/ (Accessed 22 January 2025)
Amphistar (Belgium)	Biosurfactants	WCO	https://amphistar.com/ (Accessed 22 January 2025)
Baker Commodities	Fuel, co-products for plastics, animal feed, fertilizers, soaps, cosmetics	WCO	https://bakercommodities.com/ (Accessed 22 January 2025)
Denalicop	Soaps, additive for asphalt, fertilizers	WCO	https://www.denalicorp.com/ (Accessed 23 January 2025)
Dymresource	Biodiesel, fertilizer, soaps	WCO	https://dymresources.com/ (Accessed 23 January 2025)
GF Commodities	Animal feed, biofuels, soaps, oleochemical	WCO	https://gfcommodities.com/ (Accessed 23 January 2025)
Hairma Technology	Polyols (HM-10100R)	WCO	https://www.hairma.com.cn/home/list/19 (Accessed 15 January 2025)
LyondellBasell	Polyolefins (CirculenRenew)	WCO	https://www.lyondellbasell.com/ (Accessed 23 January 2025)
MBP Solutions	Additives for asphalt	VRO ^(a)^/WCO	https://www.mbpsolutions.com/ (Accessed 23 January 2025)
Orlen	Hydrogenated vegetable oils	RSO ^(b)^/WCO	https://www.orlen.pl/en/ (Accessed 24 January 2025)
Repsol	Renewable fuels, fertilizers, candles	WCO	https://www.repsol.com/en/ (Accessed 24 January 2025)
Southern Green	Soaps, Biodiesel, animal feed additive, fertilizers, flame starters	WCO	https://www.southerngreen.com/ (Accessed 25 January 2025)
Wanhua Chemical	Polyols (Wanol^®^)	WCO	https://en.whchem.com/cmscontent/770.html (Accessed 25 January 2025)

^(a)^ VRO: virgin rape oil; ^(b)^ RSO: rape seed oil.

## Data Availability

Further material is available on request.

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
