# Peer review of "Waste Cooking Oils into High-Value Products: Where Is the Industry Going?"

_polymers, 2025, doi:10.3390/polym17070887_

Round 1

Reviewer 1 Report

Comments and Suggestions for Authors

1, "Eu"or "EU",pls check it throughout the paper. " to make thinks worse" or "to make things worse"? "percent-age point higher than"?

2, pls confirm Fig.1 is correct. In terms of WCO or cooking custom, China may produce more WCOs than USA. If Fig.1 was from reference, pls cite it.

3,   Pls make sure each acronyms explained with full name at first sight, for instance, what are "BLs"?

4, it is advised to include a figure to show the use or type of such polymeric products from WCOs. For instance, construction area, Animal feed, Fertilizers. Comments on the trend or potential values would attract readers.

5, pls check the references styles comply with the journal's requirements. Reference 138 is missing. 

Author Response

Reviewer#1
1) "Eu"or "EU",pls check it throughout the paper. " to make thinks worse" or "to make things worse"? "percent-age point higher than"?

All the paper was revised and Eu changed in EU, “think worse” to “things worse” and “percent-age point changed in “point percentage”.

2) pls confirm Fig.1 is correct. In terms of WCO or cooking custom, China may produce more WCOs than USA. If Fig.1 was from reference, pls cite it.

As suggested by reviewer 1, the reference from which data were derived has been added to the caption of Figure 1.

3) Pls make sure each acronyms explained with full name at first sight, for instance, what are "BLs"?

All acronyms were searched and full name specified at first sight. 

4) it is advised to include a figure to show the use or type of such polymeric products from WCOs. For instance, construction area, Animal feed, Fertilizers. Comments on the trend or potential values would attract readers.

In consideration of Reviewer 1 comment, Figure 3 has been added.

5) pls check the references styles comply with the journal's requirements. Reference 138 is missing.

MDPI full reference guide has been downloaded and References adjusted when necessary to the instructions.

Reviewer 2 Report

Comments and Suggestions for Authors

The review article "Waste Cooking Oils into High Value Products: Where is the Industry Going?" discusses actual processes of the conversion of waste cooking oils (WCOs) into value-added products (upcycling) taking into account realities and prospects of the industrial implementation of these processes. Two important facets of the problem are summarized and discussed in this work: patent positions (last 10 years) and promotion to the market.

The method of literature search appeared to be correct. The subject of the review is actual, the work seems interesting for a wide readership, but the article needs revision (see comments below) and doesn't quite match the aims, scope and level of the Polymers journal. Sustainability journal seems more appropriate for this work.

  1. Line 132. WCO conversion to FAMEs by alkali-catalyzed transesterification is complicated by high acidity of WCOs. Actual solutions are based on acid-catalyzed esterification before transesterification. Current approaches to FAMEs through the prism of 12 principles of Green Chemistry are discussed in many reviews, e.g. Green. Chem. 27 (2025) 41. The processes of cracking and hydrocracking of WCO results in complex mixtures of the products unrelated to FAMEs.
  2. Lines 173-178. Repetitive text
  3. Line 155 and below: this Section mainly discusses the conversion of WCOs to low-MW products. Polymers are:

– mentioned in the Table 1 (CirculenRenew)

– mentioned at pages 7 and 8 (polyols for PU, alkyd resins) and randomly and tediously listed at pages 8 and 9, without any chemical structure. It is not enough for the review in Polymers, as far as I'm concerned

  1. WCO upcycling is primarily determined by composition of WCO. No mention of this in the review.

Technical remarks:

Line 34 – For million tons, abbreviation Mt is commonly used.

Line 164 – EU commission

Line 191 – please define BLs

Line 195 – replace "more expensive" by "higher"

Author Response

Reviewer#2
The review article "Waste Cooking Oils into High Value Products: Where is the Industry Going?" discusses actual processes of the conversion of waste cooking oils (WCOs) into value-added products (upcycling) taking into account realities and prospects of the industrial implementation of these processes. Two important facets of the problem are summarized and discussed in this work: patent positions (last 10 years) and promotion to the market.

The method of literature search appeared to be correct. The subject of the review is actual, the work seems interesting for a wide readership, but the article needs revision (see comments below) and doesn't quite match the aims, scope and level of the Polymers journal. Sustainability journal seems more appropriate for this work.

1) Line 132. WCO conversion to FAMEs by alkali-catalyzed transesterification is complicated by high acidity of WCOs. Actual solutions are based on acid-catalyzed esterification before transesterification. 
Current approaches to FAMEs through the prism of 12 principles of Green Chemistry are discussed in many reviews, e.g. Green. Chem. 27 (2025) 41. The processes of cracking and hydrocracking of WCO results in complex mixtures of the products unrelated to FAMEs.

I thank the reviewer for the suggestion. Reference has been added to the text (line 146).

2) Lines 173-178. Repetitive text

As recommended by Reviewer 2 and Reviewer 3 lines 173-178 have been changed as reported below:
Additionally, the use of WCOs as feedstock for the biorefinery is struggling to find a place in the market, highly depressed in favour of the biodiesel and energy sector, which nonetheless is passing through a very critical moment in the EU. In fact, due to the strong reduction of biodiesel selling prices as a consequence of Chinese companies’ competition, in 2024 Chevron closed its biodiesel plant in Germany, BP scaled back plans for sustainable aviation fuel and renewable diesel in EU and Shell halted construction of a big biodiesel plant in Rotterdam [114]. Although the European Commission announced that provisional import duties will be imposed on Chinese biodiesel, competition for biodiesel production demonstrates that higher value added products from WCOs, prepared with innovative technologies, are the answer to support EU market competitiveness, moving away from the biodiesel market.

3) Line 155 and below: this Section mainly discusses the conversion of WCOs to low-MW products. Polymers are:

– mentioned in the Table 1 (CirculenRenew)

– mentioned at pages 7 and 8 (polyols for PU, alkyd resins) and randomly and tediously listed at pages 8 and 9, without any chemical structure. It is not enough for the review in Polymers, as far as I'm concerned

I thank the reviewer for the consideration which, yet I don’t understand what is requested. If necessary, could the reviewer please send further indications?

4) WCO upcycling is primarily determined by composition of WCO. No mention of this in the review.

Regarding this comment by Reviewer 2 I would like to point that although I’m aware of the fact that WCOs composition is of primary importance for its recycle, yet the scope of this paper is to highlight industrial processes patented and/or employed by the industry. The specific pretreatment and purification processes adopted in each case can be found, where given, in the patents and is not the focus of this paper.

Technical remarks:

Line 34 – For million tons, abbreviation Mt is commonly used.

Line 164 – EU commission

Line 191 – please define BLs

Line 195 – replace "more expensive" by "higher"

We thank reviewer 2, and all technical remarks have been added to the text

Reviewer 3 Report

Comments and Suggestions for Authors

Please, find comments and suggestions for Author enclosed as a pdf file.

Comments on the Quality of English Language

Comments on the quality of English are a part of the "Review of Polymers-Waste v1" file.

Author Response

Reviewer#3
1. Terminology:
Miot – this term as an abbreviation of a unit (million of tons) does not exist (used in Introduction, page 1, line 34, in Chapter 3, page 3 line 118, etc. and even in the conclusions). Could the Author use Mt (Megatons) or Tg (Teragrams) instead, which would be consistent with the SI, or even simpler engineering notation, i.e. t × 106?

We thank reviewer 3, and all Miot have been changed to Mt

2. Chapter 2 – Methods:
2.1. The methodology described in this chapter (whole text before the flow diagram) seems to be a standardised systematic review approach (with initial search of the scientific databases, followed by exclusion of irrelevant data and so on). The flow diagram on page 3 also resembles standardised “PRISMA” diagram. The reviewer suggests to use and describe this standardised approach to the presented data, in particular (1) to clearly formulate a database search criteria and (2) to create a flow diagram following “PRISMA” rules (see https://guides.lib.unc.edu/prisma).

I thank the reviewer for the suggestion, unfortunately I have never used this methodology, and I was not able to replicate the figure ex post. I will in any case consider Prisma as a very useful tool for future review articles.

2.2. The description of the organisation of the review (in the first paragraph of the chapter) is incoherent. Points (1) and (3) are presented as sentence equivalents (with no predicate), i.e. “selecting databases…” (line 81) and “subdivision of the articles…” (line 84), while points (2) and (4) – as proper sentences with predicates, e.g. “…review articles were first reviewed…” (line 85). The reviewer would suggest to eliminate proper sentences, e.g. (2) “analysis of titles, abstracts and conclusions…”, “…and removal of inappropriate or redundant…”

According to reviewer 3 lines 81 to 83 were rewritten as follows:
selection of database, publication type, subject areas and keywords; (2) analyses of titles, abstracts, and conclusions of the selected patents; (3) subdivision of the patents in different subsections (for example bio-surfactants, bio-lubricants). This process allowed us to identify 20,000 patents which were analysed and from this preliminary analysis only 100 patents were further taken into consideration for the present review.

2.3. The scientific databases can be searched using logical statements made of the keywords using logical operators (.AND., .OR., .NOT.). This approach can reduce number of searches, time consumed, and increase the precision (e.g. by excluding “Energy” and “Fuel”, which, as the reviewer understands, was the Author’s intention).

As mentioned above I really appreciate showing me a very interesting tool for future works allowing to reduce time and make my search much more efficient. Presently in accordance to the Reviewer comment line 92 has been modified as follows:

Logical operators (.AND., .OR., .NOT.) were used in combination of ….

2.4. Why is the first figure of the paper called “Scheme 1” (line 81)? The reviewer understands it is a flow diagram, but it still should be described as “Figure 1”.

Scheme 1 has been changed into Figure 1 and other Figures renumbered in the whole text.

2.5. Use of decimal point and comma in numbers: “20.000” (line 86) in English means 20 (twenty). If the Author means thousands, then a comma should be used as a separator – “20,000”. Besides the reviewer is not sure the Author’s intentions, as in the supplementary material the Author presented a tabular list of just 61 patents.

We thank the reviewer for the comment. The intention was 20,000 which were then read and only 100 were selected as inherent with the scope of the article (39 listed in the references and 61 in the supporting information section)

2.6. Scheme 1 (page 3, after line 101):

The flow diagram should be nominated as “Figure 1”, as already indicated above. The reviewer suggests modifying the flow diagram according to the PRISMA statement (https://guides.lib.unc.edu/prisma) as methodology presented by the Author meets exactly the Systematic Review Approach. Detailed content requires following language related modifications: “Literature type” should be replaced with “Publication type”, word “Articles” should be replaced with “Papers”, numbers should be presented using language rules (as mentioned in par. 2.5 above).

For this see answer to point 2.2

3. Chapter 3 – WCOs production and market:
3.1. The interesting information presented in the first two paragraphs of Chapter 3 (page 3-4, before the figure): The reviewer acknowledges the challenge of locating relevant and precise data and believes that the data found by the Author deserves better exposure. The reviewer proposes that the Author present the data in tabular form, incorporating the following columns (as an example):

The last row of the table should summarise the global data. The table will help the reader understand the scale of the problem globally and locally.

I thank the reviewer for the suggestion. Thanks to data found in https://cleanfuels.org/wp-content/uploads/GlobalData_UCO-Supply-Outlook_Sep2023.pdf (which has been added to the text) I was able do add the following table (Lines 133-142):

Additional values on the amount of WCOs recycled was found in the literature and data are summarised in Table 1 [73]. 
Table 1. Commercially available products from VOs and WCOs.
N°    Region    Annual WCO produced    Annual Mt WCO Recycled
        Mt    (%)(a)    Mt    (%)
1    USA    15.0    32    2.7    18
2    ASIA (China, India, Malaysia)    24.5    49    7.6    31
3    EU    4.0    6    1.0    25
4    Rest of the World    6.5    13    2.8    43
Total        50.0    14.1    28
a) Calculated as the percentage of a Region Mt production divided by Total WCO Mt yearly production worldwide (50 Mt). a) Calculated as the percentage of Mt total recycled WCOs (14.1 Mt) divided by Total WCO Mt yearly production worldwide (50 Mt).

From these data it is interesting to note that the “rest of the world” and Asia have the highest percentages of recycled WCOs which is prevalently used for bidiesel pro-duction and exported to other countries [73].

3.2. Figure 1 (Geographic distribution…) – page 4.
It is a good drawing, but unacceptable in its current form: the legend completely obscures the reference to China and partly the diagram itself. At least this is what the reviewer was given:
Please modify. Besides the figure should be signed “Figure 2” not 1 (as Figure 1 is Scheme 1).

We thank the reviewer for the comment. Figure 1 has been renumbered to Figure 2 and has been completely changed. 

3.3. The paragraph beginning "In this context, recently approved by the EU..." (page 4, lines 145 et seq.) appropriately justifies the reprioritisation of the use of WCOs for the development of high-value products over biodiesel production, but only based on EU regulations, while the EU accounts for only 6% of global WCOs production. It is recommended that the Author provide commentary on the regulatory situation and practical approaches in the much larger players in the WCOs market, namely the US, China and India, with a special interest in the US (the world's largest producer of WCOs), especially in the context of the recent politically driven retreat from taking environmental issues seriously.

In consideration of the reviewer comment a chapter on WCO regulatory situation has been added (lines 167-271):

4. WCO regulatory situation
Regarding the EU, efforts to regulate waste oils date back to the 1970s, beginning with Directive 75/439/EEC, which focused on the collection of used oils, minimizing environmental risks, and encouraging recovery and recycling technologies [43,97]. Over the past decade, the EU has introduced several regulatory revisions addressing the classification, storage, recovery, and disposal of WCOs originating from various sectors, including hospitality, households, and industry. The European Commission Decision 2014/955/EU, amending Directive 2008/98/EC and 2000/532/EC, provides a harmonized waste classification list, which is periodically updated to classify different wastes [98]. However, it is important to note that the inclusion of a waste in this list does not automatically classify a material as waste; a substance is considered as a waste only when it meets the definition outlined in Article 1(a) of Directive 75/442/EEC [99].
WCOs, categorized under European Waste Code (EWC) 200125, are not directly hazardous to human health, yet improper disposal poses environmental risks. To address these concerns, Article 1 of Directive 2008/98/EC establishes measures aimed at protecting both the environment and public health by reducing the negative effects of improper waste management and enhancing efficiency in reuse [100] Consequently, Member States are required to develop and implement waste management plans. However, balancing a unified EU policy with Member State sovereignty presents challenges, particularly in harmonizing waste management, recycling, and disposal strategies across borders [36,95].
By 2018, numerous Member States had yet to establish adequate waste management infrastructure. In response, Directive 2018/851 was introduced to amend Directive 2008/98/EC, aiming to prevent the overdevelopment of waste treatment facilities and optimize resource use [101]. This directive also aligns with the EU’s broader goal of reducing dependency on raw material imports and advancing sustainable material management through a circular economy model. It establishes the legislative framework necessary to support the EU Action Plan and Sustainable Development Goals outlined in the 2030 Agenda, with an overarching goal of achieving carbon neutrality by 2050 [31,44,102,103] A consistent and comprehensive legislative approach across the EU is essential to foster near-zero waste processes.
The End-of-Waste Directive 2008/98 represents a key milestone in this regard, de-fining criteria for converting waste into valuable secondary raw materials [101,104]. According to Directive 2008/98/EC, “recovery” refers to processes where waste is used to replace virgin materials that would otherwise be used, either within industrial operations or more broadly in the economy [104]. Meanwhile, “recycling” is described as any recovery operation in which waste materials are processed into new products, materials, or substances for their original or alternative purposes, excluding energy recovery or backfilling operations. To achieve “End-of-Waste” status, materials must meet specific conditions: they must be useful for a particular purpose, there must have an existing market or demand, they must meet applicable legislative and technical standards, and their use must not result in negative environmental or human health impacts [104].
In the United States, regulatory measures regarding waste oils have also evolved over time. In October 1997, the Environmental Protection Agency (EPA) declined a petition from several agricultural trade organizations requesting relaxed spill response regulations for facilities storing vegetable oils and animal fats under the Facility Response Plan (FRP) rule [104]. Subsequently, in 2000, the EPA revised the FRP rule to clarify its applicability to facilities that manage, store, or transport significant quantities of animal fats and vegetable oils, particularly those handling large-scale transfers over water or storing at least one million gallons of oil. This revision was aligned with the Edible Oil Regulatory Reform Act, distinguishing animal fats and vegetable oils from other oils based on their properties and environmental impacts [105].
The updated rule introduced a methodology for estimating worst-case discharge planning volumes for these oils, using principles similar to those applied to petroleum oils while incorporating tailored factors for recovery estimation in water and on land. Additionally, separate regulatory sections were designated for animal fats and vegetable oils, while maintaining the original response planning categories for small, medium, and worst-case discharges. The rule also introduced new definitions and a classification system dividing oils into Groups A, B, and C according to their specific gravity.
Further regulatory changes occurred in 2002 with the revision of the Spill Prevention, Control, and Countermeasure (SPCC) rule, which was updated to reflect the requirements of the Edible Oil Regulatory Reform Act [106]. The revised rule was structured into subparts: Subpart A outlined general requirements for all facilities, Subpart B addressed petroleum and non-petroleum oils (excluding animal fats and vegetable oils), and Subpart C detailed regulations for animal fats, greases, fish and marine mammal oils, and vegetable oils derived from seeds, nuts, fruits, and kernels. This re-vision also highlighted the environmental impact of oil spills on soil.
To align with the Circular Economy Action Plan, the establishment of a comprehensive legislative framework is essential for advancing sustainable practices, encouraging waste upcycling, and fostering near-zero waste processes [31,43,44].
The Clean Water Act also plays a crucial role in managing waste disposal, including WCO, to prevent pollution [107]. 
However, recent political trends have seen a relaxation of certain environmental regulations, leading to concerns about increased waste and pollution. Despite this, many states have implemented their own regulations and incentives for biodiesel production from WCO, promoting a circular economy approach [108].
China has seen a significant increase in the use of WCO for biodiesel production, driven by government policies aimed at reducing waste and promoting renewable energy. The National Development and Reform Commission (NDRC) and the Ministry of Ecology and Environment (MEE) oversee regulations regarding waste management and environmental protection. Very recently the Chinese State Council has issued guidelines for expediting the establishment of a waste recycling system [109]. According to the document, the construction of waste recycling systems is considered crucial to ensure the security of national resources, actively and steadily promoting carbon neutrality, and accelerating the green transformation. It established that by 2025, a waste recycling system covering all fields and links will be initially set and by 2030, a comprehensive, efficient, regulated and well-organized waste recycling system will be built up, to valorise various waste, and push overall level of waste recycling to top rank in the world.
Efforts should also be made to improve the level of recycling and reuse of waste: to strengthen the comprehensive utilization of bulk solid waste, to enhance the efficient use of renewable resources.
Meanwhile, the resource recycling industry will be fostered and improved, according to the guidelines [109]. However, challenges remain, including illegal disposal practices and a lack of stringent enforcement. The government’s recent focus on eco-nomic growth may overshadow environmental concerns, impacting WCOs initiatives.
India is experiencing a growing interest in converting WCOs mainly focusing on biodiesel production, supported by the National Biofuel Policy [110]. The regulatory framework is still evolving, with the Ministry of Petroleum and Natural Gas leading efforts to promote the use of biodiesel. As an example, in December 2023, the government of India started a campaign to promote WCOs for the purposes of biodiesel pro-duction [111]. and State owned oil companies have pledged to buy biodiesel made from UCO over the next three years. The goals were aimed to mitigate health hazards, environmental destruction, and reducing the cost of oil imports. However, enforcement of regulations can be inconsistent, and the market is often hindered by a lack of infrastructure for collection and processing. Political factors, including economic priorities, can lead to insufficient attention to environmental issues related to WCOs.

4. Chapter 4 – Where is the industry going?
4.1. The paragraph starting with: “To make things worst…” (page 5) contains nearly wordto-word copy of a statement regarding EU regulatory and its consequences written earlier in Chapter 3. Compare texts in lines 173-180 (page 5) with lines 145-152 (page 4). Avoid unnecessary repetitions, please.

As suggested by the Reviewer lines 290-305 have been modified as follows:
Additionally, the use of WCOs as feedstock for the biorefinery is struggling to find a place in the market, highly depressed in favour of the biodiesel and energy sector, which nonetheless is passing through a very critical moment in the EU. In fact, due to the strong reduction of biodiesel selling prices as a consequence of Chinese companies’ competition, in 2024 Chevron closed its biodiesel plant in Germany, BP scaled back plans for sustainable aviation fuel and renewable diesel in EU and Shell halted construction of a big biodiesel plant in Rotterdam [114]. Although the European Commission announced that provisional import duties will be imposed on Chinese biodiesel, competition for biodiesel production demonstrates that higher value added products from WCOs, prepared with innovative technologies, are the answer to support EU market competitiveness, moving away from the biodiesel market.

4.2. The paragraph starting with: “Different BLs are commercially available…” (page 5, line 191) – what are BLs? Perhaps biolubricants (?), but it has not been introduced in the paper as abbreviation. Please, add clarification.

I apologize for the inconvenience, yes BL stands for biolubricants which has been added when acronym was first used (Line 314).

4.3. The same paragraph, the value of biolubricants (if BLs means biolubricants) market is given in the text as “2.8 M$ in 2031” (line 195). The reviewer has got a report (https://www.grandviewresearch.com/industry-analysis/biolubricants-industry) giving biolubricants market value in 2024 as US$ 2.95 billion with a projection to grow by further 13.7 % between 2025 and 2030. So “M” in “M$” cannot be “million”, can it? Besides, please follow the rules of presenting money values in English (currency symbol always precedes the value).

As recommended by the reviewer line 317-319 has been changed to:
Although BL market reached US$ 2.95 billion in 2024, and is expected to grow at a compound annual rate of growth (CAGR) of 13.7% from 2025 to 2030 [117],
And suggested reference added

4.4. The paragraph starting with: “As far as BS from WCOs are concerned…” (page 6, lines 210 and following) – what are BSs? Perhaps biosurfactants or maybe biosolvents? The abbreviation has not been introduced in the paper properly. Please, add clarification.

BS stands for bio surfactants and full name has been added where first mentioned (Line 334)

4.5. The paragraph starting with: “As regards polyols…” (page 7, lines 236-247) – what are PUs? Perhaps polyurethanes? The abbreviation has not been introduced in the paper properly. Please, add clarification.

Specification that PU stands for polyurethane has been added when first mentioned.

4.6. The paragraph starting with: “Concerning NIPUs…” (page 8, lines 258-262) – what are NIPUs? Perhaps non-isocyanate polyurethanes? The abbreviation has not been introduced in the paper properly. Please, add clarification.

Specification that NIPUs stands for Non Isocyanates Polyurethanes has been added when first mentioned.

4.7. The same paragraph, last sentence (page 8, lines 261-262) – the Author mentions "Figure 4b," which evidently does not exist in the paper under review. It is possible that Figure 4b does exist in the cited publication, but the Author provides a double reference ([120,121]) in the disputed sentence. This requires clarification.

I apologise for the mistake. I initially wanted to add a figure which was finally not added. Strategies mentioned in the text are those reported in ref 120, 121

4.8. The paragraph starting with: “As far as acrylic polymers, alkyd resins…” (page 8, line 263 and following) – the Author lists acrylic polymers and alkyd resins next to each other, though they are totally different compounds with acrylic polymers being waterbased so not connected to vegetable or wasted cooking oils. What is the intention of this link?

There is actually no connection and since acrylic polymers were not further mentioned the sentence was changed in: 

As far as alkyd resins are …..

4.9. The paragraph starting with: “Regarding the industrial panorama…”, in the second sentence there is an abbreviation RAP/WCO (page 8, lines 284, 286, and 289) – what is RAP? Perhaps Reclaimed Asphalt Pavement? The abbreviation has not been introduced in the paper properly. Please, add clarification.

Again I regret not adding the acronyms in the paper. Yes RAP stands for reclaimed asphalt pavement and specification has been added in the text when first mentioned.

4.10. The same paragraph, lines 286-290: The Author writes that while the use of WCO in asphalt recycling makes the process more straightforward and economical (even citing the amount of savings), it provides a less durable product (more cracks!). The Author is invited to provide a more detailed commentary on this issue, including a quantitative assessment of the gains and losses, in light of the potential need for more frequent repairs.

I do understand that the two sentences are in contrast, yet this is exactly what the authors state. I tried to rewrite the paragraph in order to smooth the contradiction, (lines 412-415):

nonetheless, according to the authors, the protocol reported is very easy, potentially applicable at large scale and, economical studies further evidenced a significant reduction in construction costs when RAP/WCOs are employed, with up to 34–40% savings. 

4.11. The paragraph starting with “Further information on the availability…” (page 8, lines 291 and following). The last sentence of this paragraph suggests that the author found 3 more manufacturers of bio-based asphalt additives in the EU. However, these findings are not presented in the paper itself (apart from this one short note). Why? 
Could the Author please provide further information about these manufacturers, or offer an explanation as to why they have not been included in the paper?

No further information was available on the other suppliers and therefore the sentence was delated.

4.12. Page 8, the paragraph starting with “Oilstones, a company from South Korea…” (line 302) – the Korean company name is in fact OILSTONE, not Oilstones. Please modify.

Correction has been made

4.13. Page 8, the same paragraph, second sentence (lines 305-306) – the Korean company name is Hwa Shin Bolt but it produces mechanical seals and special fasteners, and does not construct roads. 

According to the reviewer comment patents (KR100472090B1, 2002, KR20040025057 A Inventor KIM SEOK JUN and  LEE HUI YEON) were checked and Assignee corresponds to WASHIN ROAD SEAL

4.14. Page 10, last sentence of the paragraph preceding Conclusions: The Author here makes reference to supplementary material that has been reviewed by the reviewer. This supplementary text file is a table containing just 61 patents from various countries (US, China, Japan, Korea etc.), with different statuses (alive, pending, revoked, expired etc.), listed in order of subject matter (biolubricants, biosurfactants, polyurethane, acrylic polymers etc.). The reviewer has previously expressed reservations regarding the number of patents, noting that the sole numerical value cited in the primary text is 20,000 patents identified during the database search. However, it remains unclear why, from these 20,000 patents, the Author selected just 61 to be included in the supplementary material. The reviewer contends that supplementary material of value should comprise a register of contemporary patents, whether currently active or pending, that present a promising outlook for the translation of patented inventions into practical applications.

I thank reviewer for the comment and in this respect the answer has already been given at point 2.2 above.

5. Chapter 5 – Conclusions:
5.1. The paragraph starting with “Within this panorama…” – this entire paragraph should be removed from this section and placed elsewhere. Its content is of significant value and relevance as it provides substantial support for the Author's thesis on the contraction of the biodiesel market. However, it is not appropriate for inclusion in the conclusions section. 

According to reviewer’s comment this section has been moved to lines 290-300.

5.2. The paragraph starting with “Data reported…” – The Author is absolutely right in this part of the conclusions, but the recommendations outlined here, e.g. “To overcome these problems stronger legislative strategies…” (lines 391 et seq.), are limited only to the EU law and market, which generates only 6% of global WCOs (as the Author rightly points out). The Author should emphasise more strongly the global responsibility in this area, particularly given that the patents listed and discussed in this paper for the technologies under consideration originate from various countries worldwide, including major WCOs generating countries.

Please see answer to point 3.3

6. Editorials and language:
• Title: is “…High Value Products” – should be “…High-Value Products” because in this case “high-value” plays a role of adjective describing products (high-value products, but products of high value).

• Abstract – the sentence: “Many reviews have been reported in the literature reporting potential WCOs applications to produce bio-lubricants…” (lines 19-22) needs rephrasing to avoid repeating the same word – in this particular sentence the word based on “report” is repeated three times (reported – reporting – reported). They could be replaced, e.g. published – reporting – presented.
• In the same sentence (line 20) – is: “…bio-lubricants, bio-solvents…” – should be
“…biolubricants, biosolvents…” (as in “biodiesel”).
• Introduction – Very similar sentence to the one mentioned above appears here (page 2, lines 69 and following) with triple use of the virtually same word (reported – reporting – reported). The reviewer would suggest replacements as above.
• In the same sentence (line 70) – is “…bio-lubricants, bio-solvents…” – should be “…biolubricants, biosolvents…”
• Page 2, line 73 is “…has been ever reported” – should be “…has ever been reported”.
• Methods: page 2, line 82 is “literature type” – should be “publication type”.
• Page 2, lines 84 and 85 is “articles” – should be “papers”.
• Page 2, lines 84-85 is “…bio-surfactants, bio-lubricants…” – should be “…biosurfactants, biolubricants…”
• Page 2, line 85 poor wording (review papers reviewed) – I would suggest “…initial analysis of review papers together with references…”
• Page 2, line 86 is “…allowed us to identify…” – should be “…allowed to identify…”
• Page 3, line 93 is “…bio lubricant, bio surfactant, bio solvent, … , bio polymer…” –
should be “…biolubricant, biosurfactant, biosolvent, … , biopolymer…”. However, this remark is valid for the publication, while in entering the keywords to the database search, the reviewer would more open for different spellings.
• Page 3, lines 100-101 (just before Sch.1) – the last sentence in this paragraph should be changed to avoid writing in the first person. Modify this sentence like this (for example): Once relevant sources were identified, references were reviewed…
• Page 3, lines 105-106 (just before WCOs production and market) – this last sentence of the paragraph is awkward – replace it with: “A further exclusion concerns enzymatic degradation processes of WCOs to produce low molecular weight compounds.”
• WCOs production and market – page 4, lines 153-154 (the last sentence before Chapter 4) – is: “…into the application of WCOs for bio-diesel production, focusing instead on alternative and innovative uses for these materials” – should be: “…into the biodiesel production from WCOs, focusing instead on alternative and more innovative applications of these materials”.
• Where is the industry going? – page 4, line 158 – is: “…WCOs uses for bio-diesel production…” – should be: “…WCOs used for biodiesel production…”
• Page 5, line 181 (first sentence of par. starting with “This said, presently…”) – too many words “presently” used in one sentence. Please modify.
• Page 5, line 182, same sentence as above – is: “…as will further be reported below…”, should be: “…as is reported below…”. But the reviewer would suggest the Author to avoid referring to this part of the text which was not read yet by the reader.
• Page 5, line 185 – use of the word “depressing” is misleading in this context. The reviewer suggests to modify this sentence for example as follows: “This is of course strictly connected to market demand and economical sustainability, with a consequent reduction in more innovative but costly alternatives to WCO valorisation.”
• Page 5, line 191 (first sentence of par. starting with “Different BLs are commercially…) – first word “are” of the two used in this sentence is to be removed (“Different BLs commercially available on the market are produced…”)
• Page 5, lines 199-201: The last sentence of the same paragraph (“This is in fact in line with reports…”) is difficult to understand. The reviewer suggest to modify as follows (for example): “This finding is consistent with reports from the biolubricant industry, which is seeking alternative raw materials to VOs in synthetic esters and polyethylene glycols, while WCOs are not even included in the list (Marker Research)”.
• Page 5, lines 204-206: The paragraph starting with “Recently, King Abdullah
University…” reference no. [100] follows number [98] – there is no number [99] in the text.
• Page 6, line 209 (end of the above mentioned paragraph) – is: “bio based” – should be: “bio-based” (as earlier, in line 186).
• Page 6, Table 1, many cases when “bio lubricants”, “solvents”, “bio surfactants”, “bio diesel” need to be replaced with “biolubricants”, biosolvents”, biosurfactants”, “biodiesel”, respectively.
• Page 7, line 217 (first paragraph after the table) – bacteria names should be in italic (Pseudomonas aeruginosa).
• Page 7, line 219 and following – the first sentence in this paragraph should be changed to avoid writing in the first person. Apart from this, the rest of the sentence is a bit awkward. The reviewer suggests to modify this sentence as follows (for example): “To the best of Author’s knowledge Amphistar is the first start-up company to sell BS products from agro-food waste, including WCOs, using the highly productive yeast Starmerella bombicola to produce more than 25 glycolipid biosurfactants.” Note that the species name has to be in italic.
• Page 7, line 236 – the paragraph starting with “As regards polyols and epoxides…” this first sentence needs modifying as follows (for example): “With regard to polyols and epoxides, the manufacturing from VOs is commercialised by many companies, such as: PolyLabs, Evonik…” Generally this original sentence is much too long – the reviewer suggests to put full stop after the references {110,111] and continue the argument in a new sentence (“Some WCOs-derived polyols are gradually…”
• Page 7, lines 254-255 (last paragraph of page 7) – is: “…environment-friendly thermosetting materials have been patented [117]” – should be: “…materials were patented [117]” – simple past would be more appropriate.
• Page 8, lines 263-264 – the first sentence of the paragraph starting with “As far as acrylic polymers…” needs modifying. The reviewer suggests the following: “With regard to the production of acrylic polymers and alkyd resins, it is evident that all commercialised products were produced from VOs, and none from WCOs”.
• Page 8, line 267 – second part of the sentence containing reference [122] includes mistakenly the phrase : “…in combination of combination of…” Please adjust. • Page 8, line 272 – the sentence on properties of alkyd resin ends with: “…and devoid harmful substances”, which needs modifying to: “…and devoid of any harmful substances”.
• Page 8, the paragraph starting with: “US patent US8895689 B2 by Valspar Sourcing, Inc.,…” (lines 277-281) needs re-phrasing as a whole. The reviewer proposes the modification below, however the Author needs to confirm her intentions are remaining in the proposal: “The US patent US8895689 B2, owned by Valspar Sourcing, Inc., Minneapolis, MN (US) (earliest priority date 2006, granted alive), describes a method, consistent with methodologies known from the literature, for producing alkyd resins using WCOs containing at least 4% unsaturated fatty acids, alongside polyols and polycarboxylic acids, which react to form alkyd resins [124].”
• Page 8, first sentence of the paragraph starting with “Regarding the industrial panorama…” (lines 282-283) needs re-phrasing as follows (for example): “Regarding the panorama of WCO applications for asphalt mixtures, there are very few examples of industrial-scale applications.”
• Page 8, the same paragraph, lines 286-287 – is: “…higher longitudinal and transverse cracks” – should be: “…higher density of longitudinal and transverse cracks”.
• Page 8, the paragraph starting with “Further information on the availability…” (lines 291 and following) needs re-phrasing to avoid repeating the name of the company. The reviewer would propose to modify as follows (for example): “Checking the availability of the bio-based asphalt additives indicated that just one producer could provide them, and moreover, they are not from WCOs. Specifically, DERNATAC® P105, patented by DRT - France [126], is a fully bio-based resin produced from Tall-Oil rosin, a paper mill by-product, esterified with pentaerythritol.”
• Page 8, the same paragraph, last sentence starting with “Further online search allowed…” (lines 297-299). The reviewer proposes to re-phrase this sentence as follows (e.g.): “A subsequent online investigation revealed three additional producers of biobased asphalt additives within the European Union.”
• Page 8, the paragraph starting with “Higher industrial interest…” (lines 300-301): why “higher”? Higher than what? The reviewer suggests to modify this, e.g.: “The high level of industry interest in using WCOs for PUs production is evidenced by the number of patents, as summarised below.”
• Page 8, the paragraph starting with “Oilstones, a company from South Korea…” (line 302 and following) – the whole text needs re-phrasing. The reviewer proposes to modify this text as follows (for example): “Oilstone, a South Korean company primarily engaged in the sale of asphalt additives and lubricants, has various granted and pending patents for the production of asphalt concrete recycling additives containing 15 to 25 % (w/w) of WCOs [127,128], although no specific product from WCOs is advertised on the company website (Oilstone.net). Also Hwashin and Road Seal, important Korean leaders in road construction (Hwashinbolt UK), patented two applications for the utilisation of WCOs as aggregate binders [129] and asphalt sealants [130], but the first has expired and the second has been invalidated and currently no product is available on the company website.”
• Page 9, the paragraph starting with “The Hubei University of Arts & Science…”, the sentence ending with reference [132] (line 319) – is “…road engineering construction…” – should be “…road construction engineering…” 
• Page 9, the paragraph starting with “A case of success is represented by…”, please check the spelling of the name of Swiss company, which is mentioned twice in this paragraph: first as “Lyoncell” (line 326) and second as “Lyondell” (line 331) (the Reviewer thinks the Author meant Lyondell Basell) with double “l” in both words.  
• Page 9, the paragraph starting with “Less fortune prospects seem to have…” – the first sentence (lines 341-343) of this paragraph needs re-phrasing. The reviewer proposes to modify it as follows: “A less auspicious outlook appears to be the rather large number of technologies patented by various international institutions for innovative applications of WCOs recycling, many of which expired very quickly after their presentation.”
• Page 9, the paragraph starting with “Also, other patents on apparently…” – the first sentence of this paragraph uses the phrase “such as” twice, which is awkward, and could be substituted with “e.g.” before the list of patents. 
• Conclusions – page 10, line 358, the first sentence in this paragraph should be changed to avoid writing in the first person (“…to the best of our knowledge”…) 
• Page 10, the paragraph starting with “In conclusion…” – is: “…bio-plasticizers and biopolyols…” (line 366) – should be: “…bioplasticisers and biopolyols…” (with no “-“).
• Page 10, the same paragraph – is: “…fossil based and VOs derived polyols…” (line 368) – should be: “…fossil-based and VOs-derived polyols…”
• Page 10, the paragraph starting with “Data reported…” – the first sentence (line 384) –is: “Data reported in this study further highlight that…” – should be: “Data reported in this study further highlights that…”

All of above Editorials and language issues have been corrected as suggested.

Round 2

Reviewer 2 Report

Comments and Suggestions for Authors

The authors have taken into account the comments, and I recommend to accept the manuscript.
Please correct Ref. 85:
Ivchenko, P.V.; Nifant'ev, I.E. The chemistry of oleates and related compounds in the 2020s. Green Chem. 2025, 27, 41–95. https://doi.org/10.1039/D4GC04862H.
Also please correct bibliography, the names of the authors should be separated by a semi-colon:
Surname1, A.; Surname2, B. Title of the Article